# FT3 to FT4 Conversion Ratio May Be an Independent Prognostic Factor in Pancreatic Cancer Patients

**DOI:** 10.3390/biomedicines11010077

**Published:** 2022-12-28

**Authors:** Alicja Majos, Ewa Sewerynek, Oliwia Grząsiak, Wojciech Ciesielski, Piotr Hogendorf, Jarosław Hołyński, Janusz Strzelczyk, Adam Durczyński

**Affiliations:** 1Department of General and Transplant Surgery, Medical University of Lodz, 90-419 Lodz, Poland; 2Department of Endocrine Disorders and Bone Metabolism, Medical University of Lodz, 90-419 Lodz, Poland

**Keywords:** sick euthyroid syndrome, T4 tissue conversion, pancreatic cancer, deiodinase

## Abstract

Preclinical evidence suggests that T4 can promote tumor growth while T3 can act conversely; therefore, the fT3 and fT4 concentrations should affect overall survival (OS) in cancer patients. The objective of the study was to look for an association between thyroid hormone concentrations in peripheral blood and OS in the pancreatic adenocarcinoma (PDAC) patients group. We included, retrospectively, 15 PDAC patients, without thyroid dysfunction under treatment, who underwent radical surgery, with no prior history of anticancer therapy. TSH, fT3, and fT4 concentrations were determined in blood samples taken preoperatively. We found that the fT3/fT4 ratio categorized into two groups (<0.22 vs. ≥0.22) dichotomized the study population into poor and good prognosis subgroups (log-rank *p* = 0.03; OS medians, respectively: 3 and 14 months), being a statistically significant predictor both in uni- and multivariate Cox regression analysis. We conclude that the importance of fT4 into fT3 conversion means not just its standard metabolic effects as the final products of thyroid gland activity. We hypothesize that it is linked to the progression of pancreatic malignancies, either via thyroid hormone receptors or indirectly, by interaction with cancer cells product.

## 1. Introduction

Thyroid hormones are characterized by a wide and multifarious influence on human metabolism. The thyroid gland (actuated by TSH) produces hormones, mainly T4 and small amounts of T3. Tissue deiodinases convert T4 into T3, which together with thyroxine-binding globulin, transthyretin, albumin, or apolipoprotein B100 provides proper bioavailability of those hormones [1,2]. The T4 into T3 conversion is not 100% efficient. There are three types of tissue deiodinases; biologically active T3 is one of three possible products of their activity, produced by D1 (deiodinase 1) and D2 (deiodinase 2); the others are biologically inactive: reverse T3 (rT3) and T2, produced by D3 (deiodinase 3), the main TSH/T4 inactivating enzyme [3]. Disturbance of their mutual balance results in a change of the T3 to T4 proportion, which, together with a lack of pituary–thyroid axis engaging (normal TSH), defines the euthyroid sick syndrome (ESS). In the ESS, the thyroid gland shows no signs of dysfunction, nonetheless, the levels of thyroid hormones are abnormal or close to the limits of the normal range. Practically, ESS is the opposite of the subclinical thyroid disorders group, where only TSH, but not T3 or T4 level is abnormal. [4]. It was described to occur in patients with a number of acute and chronic diseases, such as heart diseases, acute stroke, and cancer, among the patients of intensive care and geriatric units. Many studies investigated the probable link between thyroid disorders history and cancer, suggesting an interconnection between them [5,6,7,8,9,10]. Diverse mechanisms may constitute a basis for ESS development, regardless of them, it is usually considered to be associated with poor prognosis. Expression of tissue deiodinases seems to gain more impact on our understanding of cancer biology; we know that chronic inflammation, myocardial infarction, tissue repair, critical illness, and neoplasia: basal cell carcinoma, colon cancer—may cause D3 reactivation, reducing—at first intracellular—T3 level [3,11]. An extreme example may be “consumptive hypothyroidism”, where not only T3 but also TH level is affected by the massive activity of D3 in tumor tissue, usually occurring in infantile hemangioma patients [12]. Also intracellular regulation of TH concentration—on a microscale not involving systemic regulators—is a process important for cancerogenesis [13]. Interestingly, it was reported that levothyroxine users’ risk of developing pancreatic cancer (PDAC) is about 25% higher than in the general population [14]. Nevertheless, so far little is known about thyroid hormone tissue conversion in PDAC patients. To investigate it, we assumed two aims of this study: first, to describe thyroid hormones levels in the study population according to selected clinical features, putting a special emphasis on sick euthyroid syndrome frequency, and second, to check if thyroid hormones concentrations or their conversion ratio have a prognostic impact on overall survival (OS).

## 2. Material and Methods

This single-center clinical pilot study comprised patients with pancreatic malignancy (ductal adenocarcinoma, *n* = 15), without thyroid dysfunction under treatment and absence of thyroid dysfunction clinical signs, with no prior history of anticancer therapy, who underwent radical surgical treatment under general anesthesia in the Department of General and Transplant Surgery of Medical University in Lodz, Poland, in years 2016–2019. Peripheral blood samples were taken preoperatively (at admission). Age, sex, type of performed surgical procedure, grade, stage, preoperative TSH, fT3, fT4, CEA, CA-125, CA-19.9, CA-15.3 levels (from routine peripheral blood tests), and overall survival were analyzed. We referred to the following laboratory norms for peripheral blood substance concentrations: TSH 0.25–4.94 uIU/mL, fT3 1.71–3.71 pg/dL, fT4 0.7–1.48 ng/dL. FT4 concentration was given in other units than fT3; we decided to recalculate it into pg/dL aiming to make fT3/fT4 ratio’s interpretation more intuitive. As we aimed to investigate fT3 and fT4 concentrations and their ratio, we paid less attention to TSH. After careful consideration, we included 1 patient with a concentration below the normal range, as was incidentally undercovered and according to medical files this person was free of clinical signs of thyroid disease; 14 other patients fulfilled normal TSH concentration criterium. We decided to not include to Cox regression analysis: grade due to possible bias caused by a strong imbalance of the data sample and low T3 syndrome incidence, correlated with low fT3/fT4 ratio in multivariate analysis.

Statistical analysis was conducted using Statistica, 13.1,TIBCO software. Chi-square tests were used to compare nominal values. Comparisons of linear data were done using the U Mann–Whitney test and Spearman’s correlation. Kaplan–Meier method was used to calculate medians of survival. In subgroups, we assessed differences between survival curves with a log-rank test or its equivalent for multi-group comparisons. Cox proportional hazard model was used to analyze survival regression. All measured results were considered significant at the α level = 0.05. The study was approved by the Ethics Committee of the Medical University of Lodz in Poland.

## 3. Results

A total of 15 patients with resected PDAC were included in the study. The baseline characteristics of the study group are presented in Table 1. Eight patients were women (53%), and the median age was 60 years (range of 52–73 years). There were twelve Whipple’s procedures (80%) and three distal pancreatectomies (with splenectomy, 20%). We know about G1 in two patients (13%) and G2 in eleven (73%); patients were finally diagnosed with stages from IA to III; we categorized it into two classes, I–II (*n* = 11, 90% of them underwent Whipple’s procedure) and III (*n* = 4, 50% underwent Whipples’ procedure). Two patients (13%) met the criteria of low T3 syndrome (fT3 1.32/fT4 1.63; fT3 1.45/fT4 0.89). Median OS was 7.5 months (1. quartile: 1 month, 3 quartile 15.25 months), with 0.5 year survival at 67% and 1-year at 40% level.

TSH, fT3, and fT4 concentrations did not vary between subgroups determined according to clinical features (Table 2), although it is worth mentioning that fT4, fT3, and fT3/fT4 ratio levels changed gradually both in subgroups of <6 months, 6–12 months and >12 months survival as well as with survival time expressed in months, which can be described by Spearman’s rank correlation coefficients: r = 0.28, *p* = 0.313; r = −0.41, *p* = 0.130; r = 0.53, *p* = 0.042, accordingly (Figure 1). Also, the fT3 and fT3/fT4 ratio of patients with low T3 syndrome was significantly lower than others’ patients (1.39 (IQR 1.32–1.45) vs. 2.77 (2.19–3.11), *p* = 0.027; 0.12 (0.08–0.16) vs. 0.24 (0.21–0.27), *p* = 0.042). We measured the predictive value of the fT3/fT4 ratio using the receiver operating curves, taking as a state variable 0.5- and 1-year survival (Figure 2). We calculated AUC values for 0.5 year survival (AUC = 0.8; *p* = 0.022) and 1 year survival (AUC = 0.833; *p* = 0.002). For the proposed cut-off point (2.2) sensitivity reached 80% (0.5-year survival) and 67% (1-year survival), while specificity was, respectively, 80% and 100%.

The categorized FT3/fT4 ratio (<0.22 vs. ≥0.22) dichotomizes the study population into poor and good prognosis subgroups (*p* = 0.035; OS medians, respectively: 3 and 14 months, Figure 3); other tested parameters do not present statistical significance. We also performed the Cox regression, presented in detail in Table 3. FT3/fT4 ratio was the only one of the analyzed linear parameters (age, TSH, fT3, fT4) which revealed to be a strong, independent prognostic factor for OS (HR = 0.13, *p* = 0.017). We also constructed s multivariate Cox model, where fT3/fT4 ratio ≥ 0.22 was the strongest independent predictor for OS (HR = 0.01, *p* = 0.004). Other significant predictors were preoperative CA-15.3 (HR = 1.18, *p* = 0.025) and Whipple’s procedure (HR = 0.01; *p* = 0.024).

## 4. Discussion

Changes in serum thyroid hormone concentration accompanying different cancer have been widely described in world literature so far [6,7,8,9,15,16]; low T3 syndrome was described to have a negative predictive value in non-cancer diseases [17,18,19,20]. They also constitute an independent predictor of poor outcomes in malignant diseases, e.g., diffuse large B cell lymphoma, chronic lymphocytic leukemia, multiple meloma, lung cancer, or in patients undergoing brain tumor surgery [21,22,23,24,25,26]. Interestingly, in a Pan et al., study on multiple myeloma patients, low T3 syndrome incidence was statistically linked to a more advanced stage; we did not observe this in our work, probably due to the small size of the group and the fact, that all our probands underwent resection, which drastically changes OS prediction and the possibility of performing it is, among I-III stage patients, rather a function of local anatomy than tumor’s size or grade. Another study concerning thyroid hormones in gastric cancer patients was published in 2019 [9]. Although the authors did not analyze the fT3/fT4 ratio, the results seem to be in line with ours. They proved low fT3 and high fT4 concentrations to be associated with shorter survival, which may constitute (as the detailed results of these comparisons are not published) an equivalent of low fT3/fT4 ratio, as well as fT3 < 2.6 pg/mL and fT4 ≥ 1.2 ng/dL negative predictive value; calculation of fT3/fT4 ratio reveals the cut-off point: 0.22—the very same as we have adopted in our work. Similarly, Liu et al., in their work on thyroid cancer patients proved that those with high expression of fT3 and fT3/fT4 have a low risk of recurrence [25]. In our study group only two patients met the criteria of low T3 syndrome, nevertheless, our results confirm observations of shorter OS in patients in this group. Considering the issue of imbalanced thyroid hormones tissue conversion, the question of this phenomenon’s place in cancer development should be asked. The influence of TH and T3 on pancreatic cancerogenesis and cancer is not clear [26]. Proportionality of fT3 and D-dimer changes between peripheral and portal circulation in PDAC patients was reported [27]. This fact, together with the previously proven significance of increasing D-dimer levels in portal blood as a poor prognosis factor, constitutes an argument that tilts the scales to “the cause”, not ‘the sign’ side [27,28,29,30]. As cancer is a chronic disease, it is justified to look at it as a continuum of changes. Our measurements refer to the pretreatment state. We lack checkpoints from the further natural course of PDAC. Nevertheless, the dynamics of thyroid gland dysfunction in the example of Walker 256 carcinoma in an experimental rat model were described. The authors noted a continuous drop in fT4 and fT3 levels with normal TSH levels; they suggest that it may be due to the augmentation of some and depression of other biologic responses to thyroid hormones via post-receptor factors in tumor-bearing rats [31]. It suggests that to know its exact nature, the fT4 and fT3 disproportion in cancer patients should not only be explored in detail but also at different time points in the same patients. What is more, our findings speak in favor of the hypothesis by Nappi et al., that the switch in the D3-D2 deiodinase expression during tumor progression can constitute a breaking point for mutated grading of a tumor and testing its activity or expression may provide us, in the future, with clinically useful information. The use of CA-15.3 is not practiced today; its prognostic value for OS of high statistical significance was noted, for example by Napoli N et al., In our opinion, it is worth re-exploring in future studies [32].

The biggest limitation of this research is the small number of patients included. As they were recruited retrospectively, we were not able to balance the group in terms of grade and stage, which could affect the interpretation of multivariate analysis; also the information about their BMI and weight changes was lost. As this was a pilot study, we did not analyze the thyroid hormones’ bound forms, rT3 concentration, or serum protein level. In further research, all the mentioned remarks should not be bypassed.

We conclude that fT3/fT4 conversion ratio represents not just a potential prognostic marker for OS in PDAC. The value of this issue exploring is undeniable, because influencing those processes with both blocking and substituting thyroid is possible; nutritional treatment is possible as well. The phenomenon we observed was described in other cancers before; although the question of whether it is the cause or effect of cancer spread remains unclear, determining the precise answer may open a possibility of treatment for this state.

## Figures and Tables

**Figure 1 biomedicines-11-00077-f001:**
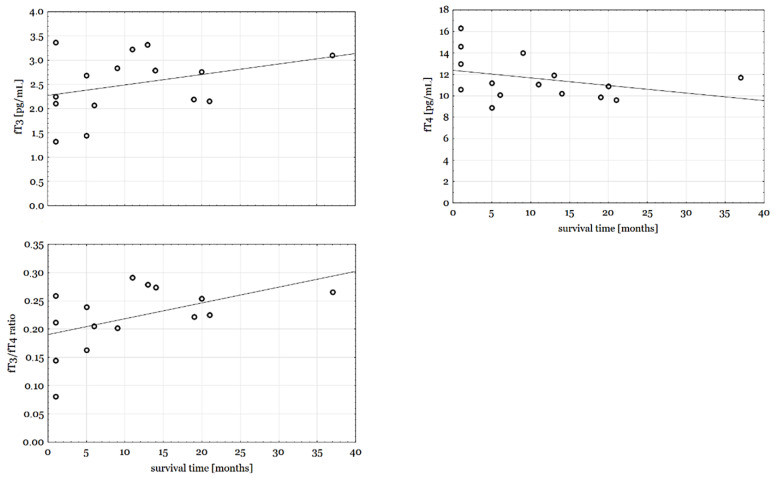
fT3, fT4 concentrations and fT3/fT4 ratio vs. overall survival.

**Figure 2 biomedicines-11-00077-f002:**
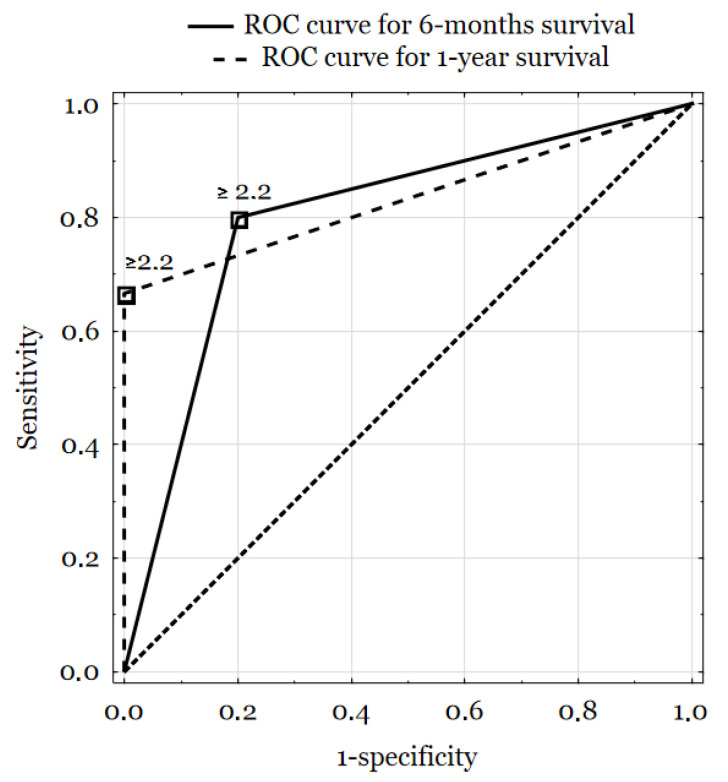
ROC curves for 6-months and 1-year survival, according to fT3/fT4 ratio status.

**Figure 3 biomedicines-11-00077-f003:**
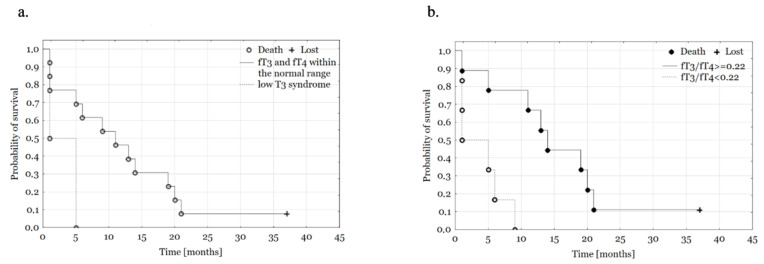
Kaplan-Meier curves in groups according to presence of low T3 syndrome (**a**) and fT3/fT4 status (**b**).

**Table 1 biomedicines-11-00077-t001:** General patient characteristics.

Thyroid Hormones Concentrations		
	Median (IQR)	Min.–Max.
Age	60 (54–67)	52–73
TSH	1.41 (0.51–2.14)	0.08–4.73
fT3 [pg/dL]	2.68 (2.11–3.11)	1.32–3.37
fT4 [pg/dL]	11.1 (10.1–13.0)	8.90–16.30
fT3/fT4 ratio	0.23 (0.20–0.27)	0.08–0.29
Clinical features; *n* (%)		
Sex, *n* (%)		
Women	8 (53.3)	
Men	7 (46.7)	
Grade		
1	2 (13.3)	
2	11 (73.3)	
Not known	2 (13.3)	
Stage		
IA	1 (6.7)	
IIA	4 (26.7)	
IIB	6 (40.0)	
III	4 (26.7)	
Type of surgery, *n* (%)		
Whipple’s procedure	12 (80)	
Distal pancreatectomy	3 (20)	
Survival, *n* (%)		
<6 months	5 (33.3)	
6–12 months	4 (26.7)	
>12 months	6 (40.0)	
Sick euthyroid syndrome, *n* (%)		
Low T3 syndrome	2 (13.3)	
fT3 and fT4 within the normal range	13 (86.7)	

**Table 2 biomedicines-11-00077-t002:** TSH, fT3, fT4 concentrations and fT3/fT4 ratio according to selected clinical features: median (IQR). *p* < 0.01: * 0.081; ** 0.027; *** 0.042.

		TSH	fT3	fT4	fT3/fT4 Ratio
Sex, (*n*, %)					
Women	8 (53)	1.79 (0.56–3.07)	2.78 (2.44–2.97)	11.05 (10.15–12.35)	0.25 (0.21–0.26)
Men	7 (47)	1.86 (0.51–2.11)	2.16 (1.45–3.23)	11.10 (9.6–14.6)	0.21 (0.14–0.28)
Stage (*n*, %)					
I–II	9 (50)	1.41 (0.51–2.14)	2.77 (2.16–3.11)	11.10 (10.20–13.00)	0.24 (0.20–0.27)
III	9 (50)	1.33 (0.32–3.04)	1.82 (1.39–2.76)	10.90 (9.40–14.10)	0.19 (0.12–0.25)
Grade, *n* (%)					
1	2 (16)	1.51 (0.87–2.14)	2.97 (2.83–3.11)	12.85 (11.70–14.00)	0.23 (0.20–0.27)
2	11 (84)	1.86 (0.51–3.97)	2.68 (2.16–3.23)	10.90 (9.90–11.90)	0.24 (0.21–0.27)
Type of surgery, *n* (%)					
Whipple’s procedure	12 (67)	1.38 (0.52–3.06)	2.47 (2.09–2.81)	11.00 (10.15–12.85)	0.22 (0.18–0.26)
Distal pancreatectomy	2 (11)	1.41 (0.08–2.11)	3.32 (2.19–3.37)	11.90 (9.90–13.00)	0.26 (0.22–0.28)
OS (*n*, %)					*
<6 months	5 (33)	1.41 (0.55–2.11)	2.11 (1.45–2.25)	13.00 (10.60–14.60)	0.16 (0.14–0.21)
6–12 months	4 (27)	1.38 (0.88–2.93)	2.76 (2.38–3.03)	11.15 (10.60–12.60)	0.22 (0.20–0.27)
>12 months	6 (40)	1.31 (0.25–2.14)	2.78 (2.19–3.11)	10.55 (9.90–11.70)	0.26 (0.23–0.27)
Sick euthyroid syndrome (*n*,%)			**		***
Low T3 syndrome	2 (13)	2.26 (0.55–3.97)	1.39 (1.32–1.45)	12.60 (8.90–16.30)	0.12 (0.08–0.16)
fT3 and fT4 within the normal range	13 (87)	1.41 (0.51–2.11)	2.77 (2.19–3.11)	11.10 (10.20–11.90)	0.24 (0.21–0.27)

**Table 3 biomedicines-11-00077-t003:** Cox proportional hazard models. Statistical significance *p* < 0.05 were signed with “*”.

**Univariate**		
	HR	*p*
Age	0.99 (0.92–1.06)	0.768
CEA	0.92 (0.78–1.09)	0.34
CA-125	1.01 (0.97–1.06)	0.622
CA-19.9	1.00 (1.00–1.00)	0.605
CA-15.3	1.02 (0.97–1.09)	0.34
Sex (male)	1.65 (0.57–4.78)	0.36
Type of procedure (Whipple’s procedure)	0.87 (0.23–3.24)	0.837
Stage (I–II)		
fT3/fT4 ≥0.22	0.71 (0.21–2.37)	0.579
	0.13 (0.03–0.70)	0.017 *
**Multivariate**		
Model		
Age		
CEA	0.93 (0.78–1.10)	0.37
CA-125	0.77 (0.57–1.06)	0.105
CA-19.9	1.00 (0.90–1.10)	0.923
CA-15.3	1.00 (1.00–1.00)	0.481
	1.18 (1.02–1.36)	0.025 *
Sex (male)		
Type of procedure (Whipple’s procedure)	3.96 (0.75–20.78)	0.104
Stage (I–II)	0.01 (0.00–0.48)	0.023 *
fT3/fT4 ≥ 0.22		
	4.06 (0.31–53.72)	0.287
	0.01 (0.00–0.25)	0.004 *

## Data Availability

The original contributions presented in the study are included in the article. Further inquiries can be directed to the corresponding author.

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
