# Peer review of "FT3 to FT4 Conversion Ratio May Be an Independent Prognostic Factor in Pancreatic Cancer Patients"

_biomedicines, 2022, doi:10.3390/biomedicines11010077_

Round 1

Reviewer 1 Report (Previous Reviewer 2)

The authors studied fT3 and fT4 level of pancreatic cancers and concluded that the level could be an independent prognostic factor.

1. This study is a retrospective analysis with small number of patients, which could have bias to the conclusion.

2. It is weird that CA153 level and type of procedure is significant in multivariate analysis, but not significant in univariate analysis (table 3).

3. In addition to type of procedure, did the patient got other therapies, such as radiation or chemotherapy,  that will impact on the survival? 

Author Response

Dear Sir or Madame,

1. There is no other possibility than to agree with you. As we did perform the study recruiting patients retrospectively, we are unable to enlarge our study group. Currently we do work on prospective version of the study, but it will definitely take time; the dilemma is to publish or to perish and we will agree, that the only one correct answer is widely known.

2. Yes, it can seem weird, but please note the HR and p for CA 15.3 in univariate analysis and compare it with HR and p levels for other markers tested (CEA, CA19.9, CEA). P value for CEA and CA15.3 are the lowest; naturally we should expect that one of them may reveal significant in multivariate analysis. Multivariate analysis is of course biased, as every kind of calculaction, by sample size, but as it takes mutual dependencies between tested parameters into account, we cannot expect its results to be ideally corresponding with those from univariate. The use of ca-15.3 is not practical - its level was marked for scientific purposes - but its prognostic value of high statistical significance was noted, for example by Napoli N et al. 

3. Our patients from the study group did not undergo preoperative chemoterapy. As we are surgical, not oncological center, our patients do not receive either postoperative chemo- or radiotherapy within the same hospital. Unfortunately it is not possible for us to gain retrospectively the specific knowledge about further course of disease. We understand the impact of this treatment on  overall survival, nevertheless, we assume that on this initial stage of investigation it is not an issue which may discredit our observation or conclusions, which we drow with caution. 

Additionally, we improved the results presentation, methods description, introduction, conclusions and citations and some english language imperfections, that we could recognize. 

Napoli N, Kauffmann E, Cacace C, Menonna F, Caramella D, Cappelli C, Campani D, Cacciato Insilla A, Vasile E, Vivaldi C, Fornaro L, Amorese G, Vistoli F, Boggi U. Factors predicting survival in patients with locally advanced pancreatic cancer undergoing pancreatectomy with arterial resection. Updates Surg. 2021 Feb;73(1):233-249. doi: 10.1007/s13304-020-00883-7. Epub 2020 Sep 25

Reviewer 2 Report (Previous Reviewer 1)

This paper is ready for publication

Author Response

Dear Sir or Madame,

We would express our gratitude and appreciation for your kind revision. 

Best regards,

Alicja Majos

This manuscript is a resubmission of an earlier submission. The following is a list of the peer review reports and author responses from that submission.

Round 1

Reviewer 1 Report

This is a clinical study associating pancreatic cancer survival and thyroid protein levels. Given that in every human cancer, staging is of paramount importance, i would urge the authors to include staging in both patient characteristics and survival results. Without staging this paper is incomplete. 

Also please include the reference for Figure 2 inside the text.

Reviewer 2 Report

The authors enrolled 18 pancreatic cancer patients retrospectively showing fT3/fT4 level could be a prognostic factor for the survival.

1. It is a retrospective study with limited number of patients, which could hamper the final conclusions.

2. Lack of staging status and treatment protocol in the analysis, which could impact on the survival analysis.

3. Lack of the pathomechanism study and/or discussion in the manuscript.